# Mumefural Ameliorates Cognitive Impairment in Chronic Cerebral Hypoperfusion via Regulating the Septohippocampal Cholinergic System and Neuroinflammation

**DOI:** 10.3390/nu11112755

**Published:** 2019-11-13

**Authors:** Jihye Bang, Min-Soo Kim, Won Kyung Jeon

**Affiliations:** 1Herbal Medicine Research Division, Korea Institute of Oriental Medicine, 1672 Yuseong-daero, Yuseong-gu, Daejeon 34054; Korea; jhbang0920@kiom.re.kr (J.B.); kms3167@kiom.re.kr (M.-S.K.); 2Convergence Research Center for Diagnosis, Treatment and Care System of Dementia, Korea Institute of Science and Technology, 5 Hwarang-ro 14-gil, Seongbuk-gu, Seoul 02792, Korea

**Keywords:** Mumefural, chronic cerebral hypoperfusion, cognitive impairment, neuroinflammation

## Abstract

Chronic cerebral hypoperfusion (CCH) causes cognitive impairment and neurogenic inflammation by reducing blood flow. We previously showed that *Fructus mume* (*F. mume*) improves cognitive impairment and inhibits neuroinflammation in a CCH rat model. One of the components of *F. mume*, Mumefural (MF), is known to improve blood flow and inhibit platelet aggregation. Whether MF affects cerebral and cognitive function remains unclear. We investigated the effects of MF on cognitive impairment and neurological function-related protein expression in the rat CCH model, established by bilateral common carotid arterial occlusion (BCCAo). Three weeks after BCCAo, MF (20, 40, or 80 mg/kg) was orally administrated once a day for 42 days. Using Morris water maze assessment, MF treatment significantly improved cognitive impairment. MF treatment also inhibited cholinergic system dysfunction, attenuated choline acetyltransferase-positive cholinergic neuron loss, and regulated cholinergic system-related protein expressions in the basal forebrain and hippocampus. MF also inhibited myelin basic protein degradation and increased the hippocampal expression of synaptic markers and cognition-related proteins. Moreover, MF reduced neuroinflammation, inhibited gliosis, and attenuated the activation of P2X7 receptor, TLR4/MyD88, NLRP3, and NF-κB. This study indicates that MF ameliorates cognitive impairment in BCCAo rats by enhancing neurological function and inhibiting neuroinflammation.

## 1. Introduction

Chronic cerebral hypoperfusion (CCH) is a major cause of vascular dementia (VaD) and can result from disorders that affect the cerebral vascular system [1,2]. The important role of CCH in dementia has emerged at the front edge of neurological studies. Accumulating evidence indicates that CCH might promote neurodegeneration through the production of reactive oxygen species and proinflammatory cytokines by activating glial, consequently leading to neuronal damage [3,4,5,6]. The main clinical feature of CCH is chronic neurodegeneration, which leads to cognitive impairment, mood disorder, impaired problem-solving ability, and loss of executive function because of impaired blood supply [7,8,9,10]. 

The bilateral common carotid artery occlusion (BCCAo) model, which is the most widely used VaD model, induces CCH by permanently ligating both common carotid arteries and shows pathological physiology similar to that in VaD patients [10,11,12,13]. We have investigated the therapeutic candidates, such as *Fructus mume* (*F. mume*), *Ginkgo biloba* L., *Salvia miltiorrhiza*, and cardiotonic pill, for CCH in the BCCAo model [14,15,16,17,18]. Notably, *F. mume*, a processed fruit of *Prunus mume*, improves BCCAo-induced cognitive impairment via the attenuation of the cholinergic system dysfunction and the inhibition of inflammation-related factors, such as gliosis, toll-like receptor-4 (TLR4), P38 mitogen-activated protein kinases (MAPK), and nuclear factor-κB (NF-κB) [15,18,19]. In addition, *F. mume* improves memory impairment through the enhanced cholinergic system in 5× FAD mice (an Alzheimer’s disease animal model) [20] and scopolamine-induced memory impairment mice [21]. The findings indicate that *F. mume* is a therapeutic candidate with excellent efficacy in improving cognition and memory. However, the active component of *F. mume* has not been identified.

Mumefural (MF) is one of the components of *F. mume* [22], fruit-juice concentrate [23], and Japanese apricot juice concentrate (*Prunus mume* Sieb. et Zucc) [24]. MF has been shown to improve human blood flow; promote erythrocyte deformability [24,25], and inhibit platelet aggregation [25]. However, it is not yet known whether MF is effective in improving cognitive function through the recovery of neurological functions. Therefore, to assess the role of CCH in VaD-related brain abnormalities and cognitive impairment, we investigated memory behaviors, the cholinergic system, myelin maintenance, synaptic protein expressions, gliosis, and NLRP3 inflammasome signaling in the BCCAo rat model. This study is the first to indicate that MF can be used as a therapeutic candidate for cerebral hypoperfusion-related neurological dysfunction. 

## 2. Materials and Methods

### 2.1. Animals

All animal procedures were performed in accordance with the protocols approved by the Institutional Animal Care and Use Committee of the Korea Institute of Science and Technology. Male Wistar rats (weight, 280 ± 10 g; age, 12 weeks; Charles River Co., Gapyung, Korea) were used in this study. Rats were housed and maintained under a 12-h light/dark cycle at 22 ± 1 °C and 55 ± 10% relative humidity, with water and chow ad libitum.

### 2.2. Brain Ischemia Surgery and Drug Treatment

Rats were anesthetized with 5% isoflurane in a mixture of 30% oxygen/70% nitrogen, and a modified BCCAo surgery was performed [14,15,18,19]. A skin incision was made to expose both the common carotid arteries, which were then separated from the vagus nerve. Both arteries were ligated with 4-0 silk sutures, and the wound was closed. Rats in the sham group underwent the same procedure without ligation. During the surgical procedure, all efforts were made to minimize pain and distress. The rats were assigned randomly into the following five groups: Sham + Vehicle, BCCAo + Vehicle, BCCAo + 20 mg/kg MF, BCCAo + 40 mg/kg MF, and BCCAo + 80 mg/kg MF. MF was purchased from U CHEM (Anyang, Gyeonggi-do, Korea). MF with the purity >95%, analyzed by HPLC, was dissolved in saline before experiments. Rats were orally administered saline or MF at 20, 40, or 80 mg/kg body weight once daily for 42 days. 

### 2.3. Morris Water Maze Task

Rats were trained and tested in a Morris water maze (MWM) [19] in order to evaluate cognition. The equipment consisted of a circular vat (180 cm in diameter, 50 cm in height), a circular platform (10 cm in diameter, 30 cm in height), and a set of photographic devices, which could record the swimming trajectory of rats. During the navigation phase, rats were trained once per day for eight consecutive days. In each round, the rats faced the pool wall and were dropped in the water gently from the midpoint of the wall edge in a random order. The swimming trajectories, time taken to find the platform (escape latency), and swimming speed were monitored and recorded. If a rat failed to find the platform within 90 s, it was guided to rest on the platform for 10 s, and its escape latency was recorded as 90 s. 

### 2.4. Immunohistochemical Staining

Animals were sacrificed at 63 days after surgery, and tissues were collected for analysis. For histological analysis, animals were transcardially perfused with normal saline, followed by 4% paraformaldehyde in 0.1 M phosphate buffered solution (PBS). After decapitation, the whole brains were post-fixed with 4% paraformaldehyde for 3 days. Subsequently, the brain tissues were dehydrated with 30% sucrose in 0.1 M phosphate buffer, embedded in Tissue-Tek^®^ O.C.T.™ Compound (Sakura Finetechnical, Tokyo, Japan), and rapidly frozen with liquid nitrogen. The brains were sectioned on a cryotome, and 40-μm sections were used for the assessment of neuronal injury. Immunohistochemical analyses of choline acetyltransferase (ChAT), myelin basic protein (MBP), ionized calcium binding adaptor molecule-1 (Iba-1), and glial fibrillary acidic protein (GFAP) were performed. Brain cryosections were prepared and incubated with primary antibodies in PBS containing 2% horse serum and 0.1% Triton-X 100 overnight at 4 °C. After washing with PBS, the tissues were incubated with anti-rabbit IgG secondary antibodies (Cell Signaling, Danvers, MA, USA). Sections were treated with a Vector SG substrate kit and a Vector DAB kit (Vector Laboratories, Burlingame, CA, USA) for peroxidase-mediated staining and were then mounted onto resin-coated slides with Permount reagent (Fisher Scientific, Pittsburgh, PA, USA). All immunoreactions were examined using light microscopy (Bx 51; Olympus, Tokyo, Japan), and the number of positive stained cell was quantified in each brain region. The number of ChAT, Iba-1, and GFAP positively stained cell body was counted. Furthermore, the densities of images were analyzed using an Image J software (NIH, Bethesda, MD, USA). The relative optical densities of MBP were expressed. A minimum of three sections was selected for each rat, and the results were averaged for analysis.

### 2.5. Western Blotting

Brain tissues were dissected and homogenized in cold lysis buffer containing 25 mM Tris HCl (pH 7.6), 150 mM NaCl, 1% nonyl phenoxypolyethoxylethanol, 1% sodium deoxycholate, 0.1% sodium dodecyl sulfate, (Thermo Scientific, MA, USA), and protease and phosphatase inhibitor cocktail solutions (GenDEPOT, TX, USA). Homogenates were centrifuged at 12,000×*g* for 30 min at 4 °C, and the supernatants were collected and stored at –70 °C until use. Protein concentrations were determined using the bicinchoninic acid assay kit (Thermo Scientific, Waltham, MA, USA), and an equivalent amount of protein (40 µg) was electrophoresed using sodium dodecyl sulfate-polyacrylamide gel electrophoresis. The proteins were transferred to a PVDF membrane, and the membrane was blocked in 5% dry non-fat milk, followed by incubation with primary antibodies. The information on primary antibodies is presented in Table 1. After incubation with primary antibodies, the membranes were washed and incubated with goat anti-rabbit horseradish peroxidase-conjugated secondary antibodies (Cell Signaling, Danvers, MA, USA). Western blots were visualized using an ECL system (Thermo Scientific, Waltham, MA, USA) with a ProteinSimple Chemi Doc machine (FluorChem E; Santa Clara, CA, USA). Band density analysis was normalized to the β-actin values with the Multi Gauge software (Fujifilm, Tokyo, Japan).

### 2.6. Enzyme-Linked Immunosorbent Assay

Brain tissues were homogenized on ice with 10 mM Tris-HCl solution (pH 7.4). The protein content was determined using a bicinchoninic acid assay kit (Thermo Fisher Scientific, Waltham, MA, USA). The acetylcholinesterase (AChE) enzymatic assay was performed using a spectrophotometer, and AChE activity was measured using a commercial kit (Sigma-Aldrich, St. Louis, MO, USA) according to the manufacturer’s instructions. In addition, levels of inflammatory cytokines such as IL-1β and IL-18 were analyzed using commercial kits (R&D, Minneapolis, MN, USA) according to the manufacturer’s instructions. All samples were examined in triplicate.

### 2.7. Statistical Analysis

Results are expressed as mean ± standard deviation (SD). The escape latency data collected from the behavioral test were analyzed using two-way repeated analysis of variance (ANOVA), followed by Tukey’s post hoc tests. Other data were analyzed using one-way ANOVA with Tukey’s post hoc tests. A *P* value of <0.05 was regarded as statistically significant. All the statistical analyses were performed using SPSS 20.0 Software (IBM, Chicago, IL, USA). 

## 3. Results

### 3.1. MF Improves BCCAo-Induced Spatial Cognitive Impairment 

We evaluated the effect of MF on cognitive impairment using MWM. Rats were trained for eight consecutive days in MWM (Figure 1A) [19]. In the navigation phase, escape latency was significantly higher in the BCCAo + Vehicle group than in the Sham + Vehicle group (*P* < 0.01), indicating that BCCAo leads to spatial learning deficits. Treatment with MF significantly decreased the escape latency compared with treatment with Vehicle (BCCAo + MF20, *P* < 0.05; BCCAo + MF40, *P* < 0.01; and BCCAo + MF80, *P* < 0.01) (Figure 1A). Moreover, the swimming speed did not significantly differ in the five groups (Figure 1B). The findings suggest that BCCAo induces cognitive impairment but does not affect motor performance. Collectively, these results indicate that MF improves cognitive impairment induced by BCCAo.

### 3.2. Effects of MF on BCCAo-Induced Cholinergic System Dysfunction 

A common feature of BCCAo rats is cholinergic system dysfunction [10,14,15,18,19]. We used ChAT immunohistochemical staining to detect cholinergic neurons in the basal forebrain (Figure 2A). One-way ANOVA revealed significant between-group effects in the regions investigated, including the medial septum (MS)/vertical limb of the diagonal band (vDB) (F(4,27) = 10.529, *P* = 0.001), horizontal limb of the diagonal band of broca (HDB) (F(4,27) = 5.663, *P* = 0.012), and nucleus basalis magnocellularis (NBM) (F(4,27) = 11.866, *P* = 0.001). As shown in Figure 2A, the number of ChAT-positive cells was significantly lower in the BCCAo + Vehicle group than in the Sham + Vehicle group in the basal forebrain sub-regions MS/vDB (*P* < 0.05), HDB (*P* < 0.05), and NBM (*P* < 0.05). The administration of MF at a dose of 40 mg/kg inhibited BCCAo-induced loss of ChAT-positive cells (MS/vDB, *P* < 0.05; HDB, *P* < 0.01; NBM, *P* < 0.05), but 20 mg/kg MF (MS/vDB, *P* < 0.05; HDB, *P* = 0.069; NBM, *P* = 0.758) and 80 mg/kg MF (MS/vDB, *P* < 0.01; HDB, *P* = 0.069; NBM, *P* < 0.01) had partial effects on BCCAo-induced loss of ChAT-positive cells. We next analyzed the expression levels of ChAT, vesicular acetylcholine transporter (VAChT), and AChE in the basal forebrain (Figure 2B). One-way ANOVA showed significant between-group effects (ChAT: F(4,15) = 12.753, *P* = 0.000; VAChT: F(4,15) = 10.534, *P* = 0.000; AChE: F(4,15) = 9.235, *P* = 0.001). Western blotting analysis revealed that MF increased ChAT expressions at doses of 20 and 40 mg/kg (*P* < 0.05), enhanced VAChT expressions at doses of 40 (*P* < 0.05) and 80 mg/kg (*P* < 0.01), and suppressed AChE expressions at doses of 40 and 80 mg/kg (*P* < 0.05). The effect of MF on AChE activity in the basal forebrain is shown in Figure 2D. One-way ANOVA revealed significant between-group effects on the AChE activity (F(4,15) = 42.565, *P* = 0.000). The increase in AChE activity due to BCCAo was significantly inhibited by MF (*P* < 0.001, Figure 2C). 

We also analyzed the protein expression levels of ChAT, VAChT, and AChE in the hippocampus (Figure 2D). One-way ANOVA showed significant between-group effects (ChAT: F(4,15) = 10.052, *P* = 0.001; VAChT: F(4,15) = 9.229, *P* = 0.002; AChE: F(4,15) = 8.530, *P* = 0.002). Western blotting analysis revealed that MF increased ChAT levels at doses of 20 and 40 mg/kg (*P* < 0.05), enhanced VAChT expressions at doses of 40 and 80 mg/kg (*P* < 0.05), and suppressed AChE expressions at doses of 40 and 80 mg/kg (*P* < 0.01). The effect of MF on AChE activity in the hippocampus is shown in Figure 2E. One-way ANOVA revealed significant between-group effects on the AChE activity (F(4,15) = 35.374, *P* = 0.000). The increase in AChE activity due to BCCAo was significantly attenuated by MF (20 mg/kg, *P* < 0.01; 40 and 80 mg/kg, *P* < 0.001). These results indicate that MF enhances the cholinergic system dysfunction in the basal forebrain and hippocampus. 

### 3.3. Effects of MF on Myelin Degradation 

The degradation of myelin, an insulating layer that forms on the myelin sheath, is a causative factor in WML damage induced by CCH; while MBP, a component in the myelin sheath, plays an important role in myelination [14,15,26,27]. We investigated the effects of MF on BCCAo-induced MBP degradation in the brain regions, such as the hippocampus, fornix, medial septum, corpus callosum, and fimbria. We quantified the density of MBP staining in the hippocampus and white matter in the BCCAo rats (Figure 3). One-way ANOVA showed significant between-group effects (CA1: F(4,27) = 4.595, *P* = 0.023; CA3: F(4,27) = 8.183, *P* = 0.003; dentate gyrus: F(4,27) = 7.654, *P* = 0.004; fornix: F(4,27) = 33.676, *P* = 0.000; medial septum: F(4,27) = 30.485, *P* = 0.000; corpus callosum: F(4,27) = 18.469, *P* = 0.000; fimbria: F(4,27) = 12.845, *P* = 0.001). The density of MBP staining in the sub-regions of the hippocampus (CA1, CA3, and dentate gyrus) was significantly lower in the BCCAo + Vehicle group than in the Sham + Vehicle group (*P* < 0.05). The administration of MF inhibited BCCAo-induced myelin degradation in the CA1, CA3, and dentate gyrus in the hippocampus (*P* < 0.05 or *P* < 0.01; Figure 3A). In addition, the density of MBP staining in the white matter was significantly lower in the BCCAo + Vehicle group than in the Sham + Vehicle group (corpus callosum and fimbria, *P* < 0.05; medial septum, *P* < 0.01). MF significantly inhibited myelin degradation induced by BCCAo in the white matter, in the medial septum, corpus callous, and fimbria (*P* < 0.05; Figure 3A). One-way ANOVA showed significant between-group effects (MBP, F(4,15) = 8.203, *P* = 0.002). Western blotting analysis revealed that MF increased MBP expressions at doses of 40 and 80 mg/kg (*P* < 0.05; Figure 3B). The findings indicate that MF attenuates MBP degradation induced by BCCAo in these brain regions. 

### 3.4. Effects of MF the Expressions of Synaptic Markers 

We investigated whether MF treatment affects the expression levels of synaptic proteins, which are involved in synaptic plasticity (Figure 4). We analyzed the expression levels of postsynaptic density protein-95 (PSD-95), synaptophysin-1, N-methyl-D-aspartate receptor (NMDAR) 2A, NMDAR2B, and phospho (p)-Ca2+/calmodulin-dependent protein kinase II (CaMKII)/CaMKII in the hippocampus. One-way ANOVA showed significant between-group effects (PSD-95: F(4,15) = 18.561, *P* = 0.000; synaptophysin-1: F(4,15) = 23.845, *P* = 0.000; NMDAR2A: F(4,15) = 7.186, *P* = 0.003; NMDAR2B: F(4,15) = 9.356, *P* = 0.001; p-CaMKII: F(4,15) = 8.530, *P* = 0.002). MF treatment at doses of 20, 40, and 80 mg/kg increased the reduced expressions of PSD-95 and synaptophysin-1 induced by BCCAo (*P* < 0.05; Figure 4A). Further, we investigated the expression levels of NMDA receptors. PSD-95 binds to and colocalizes with NMDA receptors at postsynaptic sites. MF treatment restored the reduced expression levels of NMDAR2A (20 mg/kg MF, *P* < 0.05) and NMDAR2B (20, 40, and 80 mg/kg MF; *P* < 0.05; Figure 4B) induced by BCCAo. It is known that *p*-CaMKII promotes synaptic transmission and enhances cognition [28,29]. Thus, we also quantified *p*-CaMKII expression levels. BCCAo rats exhibited significantly reduced *p*-CaMKII expression levels, but MF treatment enhanced the *p*-CaMKII levels at doses of 40 and 80 mg/kg (*P* < 0.05; Figure 4C). These results suggest that MF enhances synaptic function via the NMDAR-dependent pathway and CaMKII activation in BCCAo rats. 

### 3.5. Effects of MF on the Expression of Cognition-Related Markers 

Brain-derived neurotrophic factor (BDNF) plays a critical role in neurogenesis, synaptic plasticity, and behavior [30,31]. Hence, we investigated the effects of MF on the expression levels of BDNF and cAMP response element binding (CREB) in the hippocampus. Figure 5 showed that BCCAo decreased the protein expression levels of BDNF and *p*-CREB in the hippocampus. One-way ANOVA revealed significant between-group effects (BDNF: F(4,15) = 15.56, *P* = 0.000; *p*-CREB/CREB: F(4,15) = 9.23, *P* = 0.001). Western blotting analysis showed that MF increased BDNF expression levels at doses of 20, 40, and 80 mg/kg (*P* < 0.05) and upregulated *p*-CREB/CREB expression levels at doses of 40 and 80 mg/kg (*P* < 0.05). Therefore, MF improves BCCAo-induced decrease in the expression levels of cognition-related proteins in the hippocampus. 

### 3.6. MF Inhibits BCCAo-Induced Gliosis in the Hippocampus and White Matter

Gliosis develops around glial cells, such as microglial cells and astrocytes, when the central nerve is damaged by ischemia [32,33]. It is well known that gliosis is induced by BCCAo in the hippocampus and white matter [32,33]. Thus, we quantified the number of Iba-1-positive-microglia in the hippocampus and white matter in the BCCAo rats (Figure 6). One-way ANOVA indicated significant between-group effects (CA1: F(4,27) = 19.345, *P* = 0.000; CA3: F(4,27) = 7.711, *P* = 0.004; dentate gyrus: F(4,27) = 9.295, *P* = 0.002; corpus callosum: F(4,27) = 7.176, *P* = 0.005; fimbria: F(4,27) = 7.131, *P* = 0.006; optic tract: F(4,27) = 33.682, *P* = 0.000). The number of Iba-1-positive cells in the sub-regions in the hippocampus was significantly higher in the BCCAo + Vehicle group than in the Sham + Vehicle group (CA1 and dentate gyrus, *P* < 0.01; CA3, *P* < 0.001). Administration of MF inhibited BCCAo-induced microglial activation in the CA1, CA3, and dentate gyrus in the hippocampus (*P* < 0.05 or *P* < 0.01; Figure 6A). In addition, the number of Iba-1-positive cells in the white matter was significantly higher in the BCCAo + Vehicle group than in the Sham + Vehicle group (corpus callosum, *P* < 0.05; fimbria and optic tract, *P* < 0.01). MF significantly attenuated the increase in the number of microglial cells induced by BCCAo in the white matter in the corpus callous (*P* < 0.05), fimbria (*P* < 0.05), and optic tract (*P* < 0.01) (Figure 6B).

Moreover, we quantified the number of GFAP-positive astrocytes in the hippocampus and white matter in the BCCAo rats (Figure 7). One-way ANOVA revealed significant between-group effects (CA1: F(4,27) = 23.377, *P* = 0.000; CA3: F(4,27) = 3.586, *P* = 0.075; dentate gyrus: F(4,27) = 5.865, *P* = 0.062; corpus callosum: F(4,27) = 10.561, P = 0.001; fimbria: F(4,27) = 8.145, P = 0.002; optic tract: F(4,27) = 35.884, *P* = 0.000). The number of GFAP-positive cells was higher in the CA1 region of the hippocampus in the BCCAo + Vehicle group than in the Sham +Vehicle group (*P* < 0.01). However, MF administration attenuated BCCAo-induced increase in astrocytes in the CA1 region of the hippocampus (*P* < 0.05 or *P* < 0.01; Figure 7A). Moreover, the number of GFAP-positive cells in the white matter was significantly higher in the corpus callosum, fimbria, and optic tract in the BCCAo + Vehicle group than in the Sham + Vehicle group (*P* < 0.01). MF significantly attenuated the increase in astrocytes induced by BCCAo in the white matter in the corpus callous (*P* < 0.05 or *P* < 0.01), fimbria (*P* < 0.05), and optic tract (*P* < 0.05 or *P* < 0.01) (Figure 7B). These results indicate that MF may suppress neuroinflammation by inhibiting BCCAo-induced gliosis.

### 3.7. MF Inhibits the Activation of Neuroinflammation in BCCAo Rats

We investigated whether MF affects the inflammasome signaling, which is known to play an important role in glial-derived neuroinflammation. One-way ANOVA revealed significant between-group effects on the expression levels of P2X7 receptor (P2X7R) (F(4,15) = 11.155, *P* = 0.001), TLR4 (F(4,15) = 13.278, *P* = 0.001), and myeloid differentiation primary response 88 (MyD88) (F(4,15) = 12.271, *P* = 0.001) in the hippocampus. Treatment with MF suppressed BCCAo-related effects (Figure 8A). Western blotting analysis showed that MF decreased P2X7R expressions at doses of 80 mg/kg (*P* < 0.001) and TLR4 expressions at doses of 40 mg/kg (*P* < 0.05) and 80 mg/kg (*P* < 0.01). We further investigated the hippocampal expression of TLR4 downstream mediators, MyD88. MF treatment attenuated BCCAo-related increases in MyD88 expression (40 mg/kg, *P* < 0.05; 80 mg/kg, *P* < 0.01). Notably, MF at the dose of 20 mg/kg did not have statistical effects on the expression levels of P2X7R, TLR4, and MyD88. These results suggest that MF alleviates BCCAo-related changes through a TLR4/MyD88-dependent pathway.

Moreover, BCCAo significantly upregulated the protein levels of NLRP3 inflammasome, i.e., caspase-1 and inflammatory cytokine such as IL-1β and IL-18 (*P* < 0.05 or *P* < 0.01; Figure 8A). The activation of the NLRP3 inflammasome has two steps. The priming step includes induction of pro-IL-1β [34,35,36]. IL-1β is a pro-inflammatory cytokine derived from pro-IL-1β, and its maturation relies on the presence of caspase-1. The activated signal triggers proteolytic cleavage of dormant procaspase-1 into active caspase-1, which converts the cytokine precursors (pro-IL-1β) into mature and biologically active IL-1β. We examined the NLRP3 inflammasome pathway by Western blotting analysis. One-way ANOVA revealed significant between-group effects on the expression levels of P2X7R, TLR4, MyD88, NLRP3, caspase 1, IL-1β, and IL-18 (NLRP3: F(4,15) = 25.647, *P* = 0.000; caspase 1: F(4,15) = 8.689, *P* = 0.010; IL-1β: F(4,15) = 6.956, *P* = 0.020; IL-18: F(4,15) = 7.872, *P* = 0.012) in the hippocampus. MF treatment attenuated BCCAo-induced increases in NLRP3 expressions at doses of 20, 40, and 80 mg/kg (*P* < 0.05 or *P* < 0.01), caspase-1 expressions at doses of 40 and 80 mg/kg (*P* < 0.05), IL-1β expressions at the dose of 80 mg/kg (*P* < 0.05), and IL-18 expressions at doses of 40 and 80 mg/kg (*P* < 0.05). 

Upregulation of inflammation requires activation of the NF-κB signaling pathway; we, thus, investigated whether BCCAo activates the NF-κB pathway. As expected, BCCAo increased nucleus translocation of NF-κB subunits, p65, and p50. One-way ANOVA revealed significant between-group effects on the expression levels of p65 (F(4,15) = 9.587, *P* = 0.004) and p50 (F(4,15) = 18.505, *P* = 0.001) in the hippocampus. Notably, MF inhibited the nuclear translocation of NF-κB (p65 and p50; Figure 8B). These results suggest that MF downregulates BCCAo-associated neuroinflammation in the hippocampus. 

The signal transducer and activator of transcription 3 (STAT3) is known to be involved in the inflammatory response as a transcription factor [37,38]. The activation and interaction between STAT3 and NF-κB play an important role in controlling inflammation. One-way ANOVA revealed significant between-group effects on the expression levels of p-STAT3 (p-STAT3Tyr705: F(4,15) = 38.652, *P* = 0.000; p-STAT3Ser727: F(4,15) = 24.155, *P* = 0.000; Figure 8C) in the hippocampus. Western blotting analysis showed that MF decreased p-STAT 3 Tyr705 and p-STAT3Ser727 expressions at doses of 40 and 80 mg/kg (*P* < 0.05).

The levels of proinflammatory cytokines including IL-1β and IL-18 were significantly higher in the hippocampus in the BCCAo group than in the sham group (*P* < 0.001, Figure 8D). MF decreased IL-1β levels at doses of 40 *(P* < 0.01) and 80 mg/kg (*P* < 0.001) and IL-18 levels at doses of 20 (*P* < 0.01), 40 (*P* < 0.01), and 80 mg/kg (*P* < 0.001). 

Therefore, the findings indicate that administration of MF has mitigating effects on increased gliosis, activated NLRP3 inflammasome, and upregulated proinflammatory cytokines levels in the hippocampus in BCCAo rats.

## 4. Discussion

MF, 2-{2-[(5-formylfuran-2-yl)methoxy]-2-oxoethyl}-2-hydroxybutanedioic acid, is one of the components in Japanese apricot juice concentrate, *F. mume*, and lemon juice concentrate [22,24,39]. It is well known that MF has excellent efficacy in the improvement of human blood flow using microchannel instrument [24], inhibition of platelet aggregation [25], and antioxidant effects. Japanese apricot juice concentrates with MF reduce blood pressure in patients with grade I hypertension in a clinical study [40], improve blood flow [41], and protect against vascular hypertrophy [25]. Lemon juice concentrates with MF have an excellent antioxidant capacity [39]. In addition, our previous study found that *F. mume* improved cognitive impairment through the regulation of cerebral neurological dysfunction in animal models [15,18,19,20]. In this study, we hypothesized that MF is effective in enhancing of blood flow reduction and improving cognitive dysfunction induced by CCH. Therefore, we investigated the possible molecular mechanisms underlying CCH-induced cognitive impairment in the brains in CCH rats, and our findings indicate that MF can attenuate CCH-induced cognitive impairment. 

Long-term reduced cerebral blood flow due to CCH causes cerebral damage and cognitive impairment. Among behavior tasks, the MWM task is related to spatial and learning memory and is widely used to measure cognitive function [42]. We first assessed the time until the visit of the raised colored platform before the MWM task to examine if the rats have a visual problem [43], and we found that these animals had no any visual problems; in addition, there were no differences in body weight or swimming ability among these rats. This study found that MF administration improved BCCAo-induced cognitive impairment in a dose-dependent manner. 

Tanaka et al. reported that cholinergic dysfunction has been found to be associated with discrimination learning in BCCAo rats, suggesting that rats subjected to hypoperfusion can be used to investigate cerebral vascular dementia [44]. Cholinergic neurons, which are predominately located in the basal forebrain and project into the hippocampus and cortex, play a critical role in learning and memory functions [45,46]. We previously observed that *F. mume* increased the number of ChAT-positive stained cells in the HDB in the basal forebrain in the BCCAo model [15], and MF administration increased the number of ChAT-positive stained cells in all the three regions (HDB, MS/vDB, and NBM) with large quantities of cholinergic neurons in the basal forebrain. Hence, MF improves CCH-induced cognitive impairment by upregulating cholinergic transmission.

CCH-induced decrease in synaptic transmission is closely related to the deterioration of cognitive function [47]. Synaptic transmission involves the release of a neurotransmitter from the pre-synaptic neuron, and the neurotransmitter then binds to specific post-synaptic receptors. Synaptophysin-1, a presynaptic marker, is decreased in VaD patients [48]. Activated NMDAR2A and NMDAR2B interact with PSD-95, and the activation of CaMKII enhances synaptic transmission in the hippocampus [49]. Moreover, BDNF plays an essential role in the modulation of synaptic function, and CREB regulates the expression of the genes involved in neuroplasticity and long-term memory formation [50,51]. In addition, BDNF and CREB expressions are thought to be molecular markers for cognitive function. In this study, MF restored BCCAo-induced decrease in NMDAR, CAMKII, BDNF, and CREB protein levels in the hippocampus. The findings suggest that MF treatment improves BCCAo-induced cognitive impairment through the regulation of synaptic transmission. 

Neuroinflammation, one of the CCH-associated phenomena, induces cognitive impairment and occurs in various brain regions. We have previously demonstrated that gliosis and neuroinflammation cytokine expressions are elevated in the hippocampus in the BCCAo model [6]. CCH-induced production of IL-1β, which is mediated by the NLRP3 inflammasome activation in the hippocampus, is a key pathological mechanism underlying dementia [36,52]. NLRP3 inflammasome, a molecular marker involved in the inflammatory response, is known to play a pivotal role in the development of CCH [34,52]. The P2X7 receptor is activated by elevation of extracellular ATP, consequently leading to activation of the NLRP3 inflammasome complex [53]. In addition, Thakar et al. demonstrated that P2X7R and Iba-1 positive microglia are colocalized in the cerebral hypoperfusion-induced brain hippocampus [54]. Studies have shown that the TLR4/MyD88 signaling pathway is involved in neuroinflammation after CCH induction in BCCAo animal models [16,17,18]. In this study, interestingly, a significant reduction in these pathological changes was found in MF-treated BCCAo rats, indicating that MF suppresses BCCAo-induced gliosis in the white matter and hippocampusMoreover, MF significantly decreased expressions of P2X7R and TLR4/MyD88 and inhibited the activation of NLRP3 inflammasome, NK-κB, and STAT3 in the hippocampus in BCCAo rats. However, MF does not affect neuron density in the sub-regions of the hippocampus in BCCAo rats (Appendix A). These findings indicate that MF improves cognitive impairment through inhibiting CCH-induced gliosis, activation of NLRP3 inflammasome, and upregulation of NF-κB and STAT3.

Myelin, an insulator of neuronal axons, is known to be vulnerable to ischemic and inflammation conditions, and myelin degradation is known to be an essential cause of CCH-induced cognitive impairment [55,56]. Particularly, the corpus callosum in the white matter region is reported to be involved in the deterioration of cognitive function [57]. Our previous studies revealed myelin degradation in the corpus callosum in the BCCAo animal model [18]. Notably, this study indicated that MF attenuated BCCAo-induced myelin degradation in the white matter and hippocampus. Therefore, MF improves cognitive dysfunction probably by inhibiting demyelination caused by CCH.

Collectively, this study found that MF improved cognitive impairment and recovered cerebral function in the BCCAo model. Further studies are underway to investigate the target validation and mechanism of action of MF. 

## Figures and Tables

**Figure 1 nutrients-11-02755-f001:**
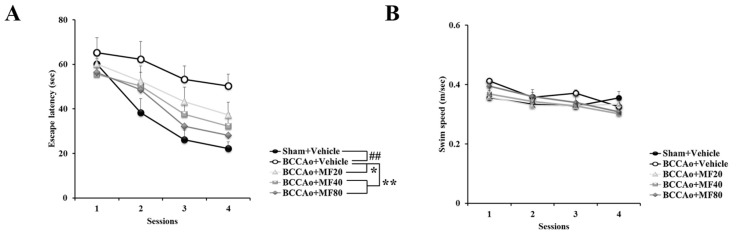
Mumefural (MF) improves bilateral common carotid artery occlusion (BCCAo)-induced spatial learning deficits and memory loss. Escape latency (**A**) and swimming speed (**B**) were assessed during the training sessions. Data are expressed as mean ± standard deviation. *## p* < 0.01, compared with the Sham + Vehicle group; ** p* < 0.05 and *** p* < 0.01, compared with the BCCAo + Vehicle group; *n* = 9–12 rats per group. MF: mumefural; BCCAo: bilateral common carotid artery occlusion; MWM: Morris water maze.

**Figure 2 nutrients-11-02755-f002:**
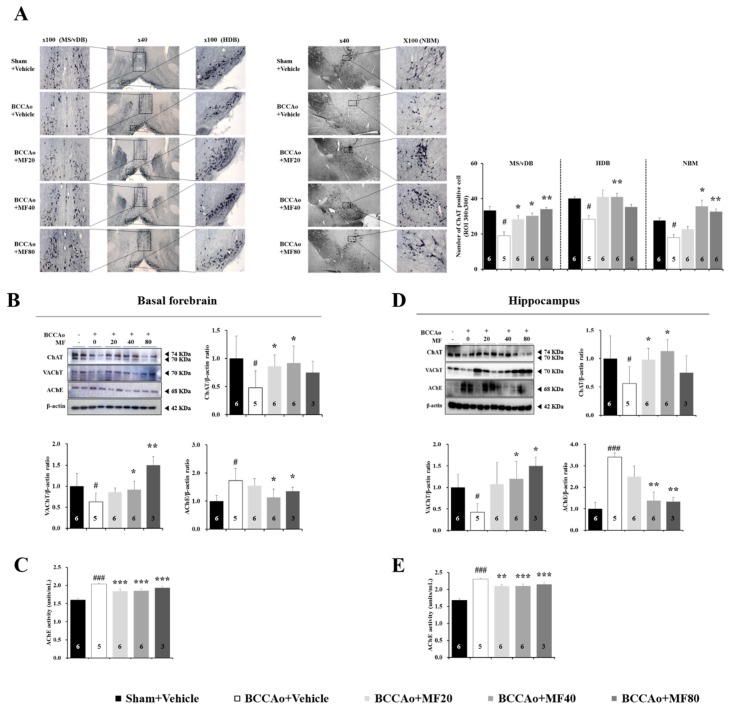
MF ameliorates BCCAo-induced cholinergic system dysfunction in the basal forebrain. Representative immunohistochemical images of ChAT-positive cells in the MS/vDB, HDB, and NBM are shown (**A**). Cells were counted under 40× magnification. Protein expression levels of ChAT, VAChT, and AChE were assessed in the basal forebrain using Western blotting (**B**). AChE activity in the basal forebrain (**C**) and protein expression levels of ChAT, VAChT, and AChE in the hippocampus (**D**) were assessed. The activity of AChE in the hippocampus was measured using ELISA (**E**). Data are expressed as mean ± standard deviation. *# p* < 0.05 and *### p* < 0.001, compared with the Sham + Vehicle group; ** p* < 0.05, *** p* < 0.01, and **** p* < 0.001, compared with the BCCAo + Vehicle group; n = 3-6 rats per group. Statistical analysis was performed using one-way analysis of variance, followed by Tukey’s post hoc test. MF: mumefural; ChAT: choline acetyltransferase; VAChT: vesicular acetylcholine transporter; AChE: acetylcholinesterase; MS/vDB: medial septum/vertical limb of the diagonal band; HDB: horizontal limb of the diagonal band of Broca; NBM: nucleus basalis magnocellularis; BCCAo: bilateral common carotid artery occlusion.

**Figure 3 nutrients-11-02755-f003:**
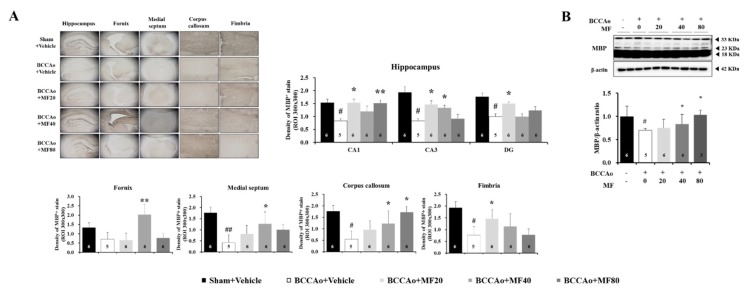
MF attenuates MBP degradation in BCCAo rats. Density levels of MBP were assessed using immunohistochemistry (**A**). Representative Western blots and graphs of densitometric analysis of MBP in the hippocampus (**B**). Data are expressed as mean ± standard deviation. *# p* < 0.05 and *## p* < 0.01, compared with the Sham + Vehicle group; ** p* < 0.05 and *** p* < 0.01, compared with the BCCAo + Vehicle group; *n* = 3-6 rats per group. Statistical analysis was performed using one-way analysis of variance, followed by Tukey’s post hoc test. MF: mumefural; MBP: myelin basic protein; BCCAo: bilateral common carotid artery occlusion.

**Figure 4 nutrients-11-02755-f004:**
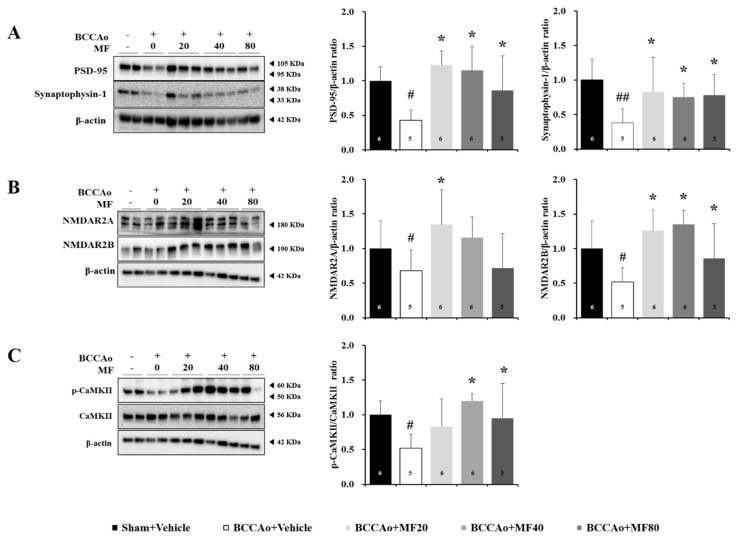
MF increases synaptic plasticity in the hippocampus in BCCAo rats. Representative Western blots and graphs of densitometric analysis of PSD-95 and synaptophysin-1 (**A**), NMDAR2A and NMDAR2B (**B**), and *p*-CaMKII (**C**) in the hippocampus. Data are expressed as mean ± standard deviation. *# p* < 0.05 and #*# p* < 0.01, compared with the Sham + Vehicle group; ** p* < 0.05, compared with the BCCAo + Vehicle group; *n* = 3–6 rats per group. Statistical analysis was performed using one-way analysis of variance, followed by Tukey’s post hoc test. MF: mumefural; PSD-95: postsynaptic density protein-95; NMDAR2A: N-methyl-D-aspartate receptor 2A; NMDAR2B: N-methyl-D-aspartate receptor; p-CaMKII: phosphorylated calcium/calmodulin-dependent protein kinase; BCCAo: bilateral common carotid artery occlusion

**Figure 5 nutrients-11-02755-f005:**
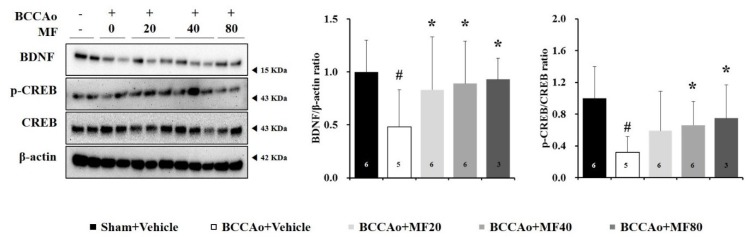
MF ameliorates impaired BDNF and CREB signaling in the hippocampus in BCCAo rats. Protein expression levels of BDNF and CREB in the hippocampus were assessed using Western blotting. Data are expressed as mean ± standard deviation. *# p* < 0.05, compared with the Sham + Vehicle group; ** p* < 0.05, compared with the BCCAo + Vehicle group; *n* = 3–6 rats per group. Statistical analysis was performed using one-way analysis of variance, followed by Tukey’s post hoc test. MF: mumefural; BDNF: brain-derived neurotrophic factor; p-CREB: phosphorylated-cAMP response element-binding protein; CREB: cAMP response element-binding protein; BCCAo: bilateral common carotid artery occlusion.

**Figure 6 nutrients-11-02755-f006:**
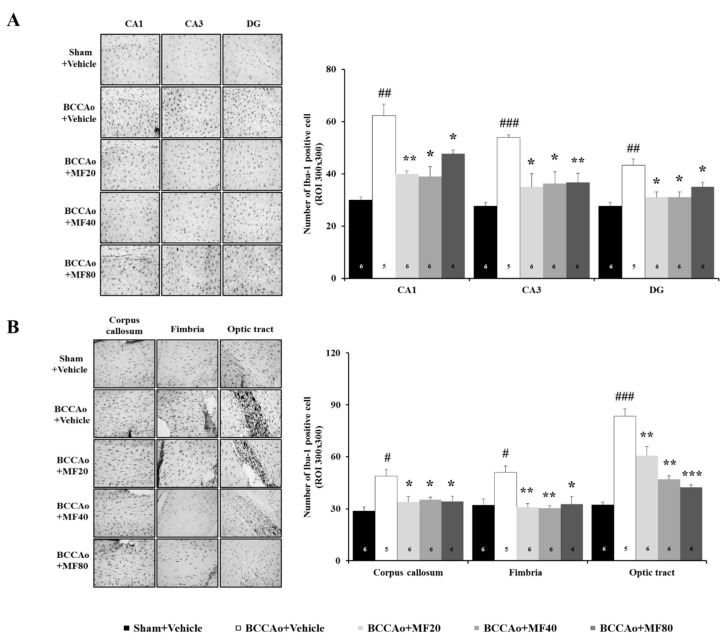
MF suppresses microgliosis in the hippocampus and white matter in BCCAo rats. Representative immunohistochemical images of Iba-1-positive cells in the hippocampus (**A**) and white matter (**B**); corpus callosum, fimbria, and optic tract) under 100× magnification. The number of Iba-1-positive microglia was counted. Data are expressed as mean ± standard deviation. *# p* < 0.05, *## p* < 0.01, and *### p* < 0.001, compared with the Sham + Vehicle group; ** p* < 0.05, *** p* < 0.01 and **** p* < 0.001 compared with the BCCAo + Vehicle group; *n* = 5–6 rats per group. Statistical analysis was performed by one-way analysis of variance, followed by Tukey’s post hoc test. MF: mumefural; Iba-1: ionized calcium-binding adapter molecule 1; BCCAo: bilateral common carotid artery occlusion.

**Figure 7 nutrients-11-02755-f007:**
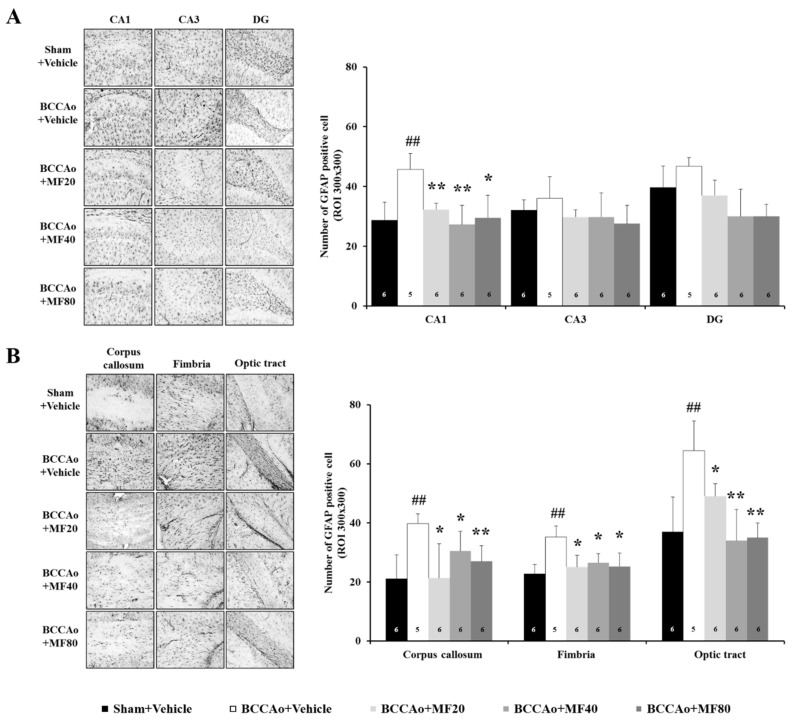
MF decreases astrogliosis in the hippocampus and white matter. Representative immunohistochemical images of GFAP-positive cells in the hippocampus (**A**) and white matter (**B**)**;** corpus callosum, fimbria, and optic tract) under 100× magnification. The number of GFAP-positive astrocytes was counted. Data are expressed as mean ± standard deviation. *## p* < 0.01, compared with the Sham + Vehicle group; ** p* < 0.05 and *** p* < 0.01, compared with the BCCAo + Vehicle group; *n* = 5–6 rats per group. Statistical analysis was performed by one-way analysis of variance, followed by Tukey’s post hoc test. MF: mumefural; GFAP: glial fibrillary acidic protein; BCCAo: bilateral common carotid artery occlusion.

**Figure 8 nutrients-11-02755-f008:**
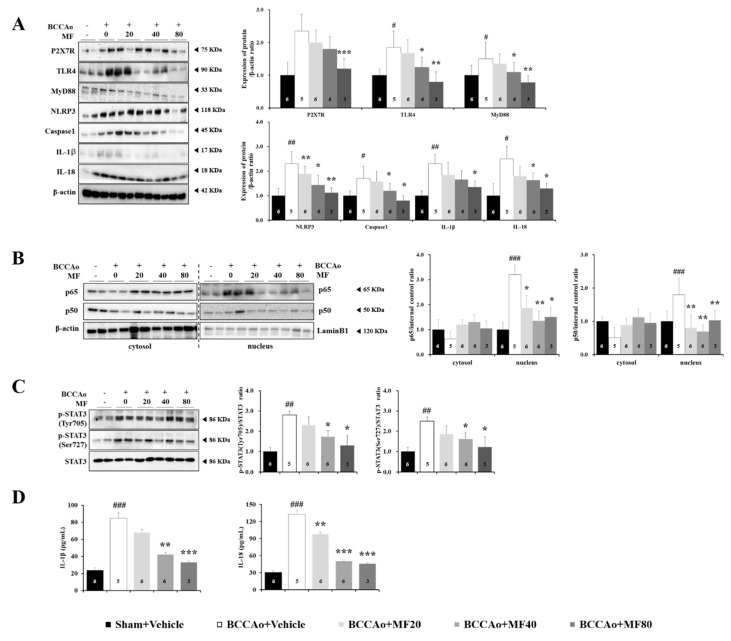
MF inhibits the activation of inflammasome signaling in the hippocampus in BCCAo rats. Protein expression levels of P2X7R, TLR4, MyD88, caspase 1, IL-1β, and IL-18 (**A**); NF-κB (**B**); p65 and p50); and p-STAT3 (**C**) in the hippocampus were assessed using Western blotting. Protein expression levels of IL-1β and IL-18 in the hippocampus were examined using ELISA (**D**). Data are expressed as mean ± standard deviation. *# p* < 0.05, *## p* < 0.01, and *### p* < 0.001, compared with the Sham + Vehicle group; ** p* < 0.05, *** p* < 0.01, and **** p* < 0.001, compared with the BCCAo + Vehicle group; *n* = 3–6 rats per group. Statistical analysis was performed using one-way analysis of variance, followed by Tukey’s post hoc test. MF: mumefural; P2X7R: P2X7 receptor; TLR4: toll-like receptor 4; IL-1β: interleukin-1β; NF-κB: nuclear factor kappa B cells; BCCAo: bilateral common carotid artery occlusion.

**Table 1 nutrients-11-02755-t001:** Details of the primary antibodies for Western blotting and immunohistochemistry.

	Antibodies	Companies	Dilution
Cholinergic System Dysfunction	ChAT	Millipore	1:500
	AChE	Abcam	1:1000
	VAChT	Millipore	1:1000
Myelin Degradation	MBP	Abcam	1:2000
Synapse Plasticity	PSD-95	Almone Labs	1:1000
	Synaptophysin-1	Almone Labs	1:1000
	p-CaMKII	Cell signaling	1:1000
	CAMKII	Cell signaling	1:1000
	NMDAR2A	Abcam	1:1000
	NMDAR2B	Abcam	1:1000
Cognitive Function	BDNF	Abcam	1:1000
	p-CREB	Cell signaling	1:1000
	CREB	Cell signaling	1:1000
Gliosis	Iba-1	Wako	1:1000
	GFAP	Sigma Aldrich	1:2000
Inflammation	P2X7R	Almone Labs	1:2000
	TLR4	Santa Cruz	1:1000
	MyD88	Santa Cruz	1:1000
	NLRP3	Abcam	1:1000
	Caspase1	Abcam	1:1000
	IL-1β	Abcam	1:1000
	IL-18	Millipore	1:1000
	p-STAT3 ^(Tyr705)^	Cell signaling	1:1000
	p-STAT3 ^(Ser727)^	Cell signaling	1:1000
	STAT3	Cell signaling	1:1000
	p-65	Santa Cruz	1:1000
	p-50	Santa Cruz	1:1000
Internal Controls	Lamin B1	Sigma Aldrich	1:500
	β-actin	Sigma Aldrich	1:2000

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
