# Peer review of "Mumefural Ameliorates Cognitive Impairment in Chronic Cerebral Hypoperfusion via Regulating the Septohippocampal Cholinergic System and Neuroinflammation"

_nutrients, 2019, doi:10.3390/nu11112755_

Round 1
Reviewer 1 Report
General comments:
This is an excellent work of utmost interdisciplinary interest and importance to the field.
Unfortunately, I do not have suggestions for further improvements of the manuscript except the questions and the minor suggestions for changes as described below in specific comments.
Please check your references and correct and complete them as requested in the instruction to authors.
Specific comments
Line 12 and 13: correct format and type size?
Line 40: bold?
Line 60: bold?
Page 4: Figure 1 difficult to read.
Page 6: Figure 2 difficult to read.
Line 209 to 226: Fomat in italics correct?
Page 9: Figure 3 difficult to read.
Page 11: Figure 4 difficult to read.
Page 16: Figure 6 difficult to read.
Page 17: Figure 7 difficult to read.
Page 19: Figure 8 difficult to read.
Line 404: Tanaka et al. reported
Line 405: that rats subjected to hypoperfusion
Line 432: Thaker et al.
References
Pleas change format as requested and provide all additional necessary information as needed.
Author Response
We wish to resubmit an article for publication in Nutrients, titled “Mumefural ameliorates cognitive impairment in chronic cerebral hypoperfusion via regulating the septohippocampal cholinergic system and neuroinflammation.” The manuscript ID is nutrients-636993.
We appreciate the valuable comments and suggestions from the editors and the reviewers, which have helped us to improve the manuscript. We have made every effort to address the concerns raised, and we feel that the paper has significantly improved as a result. Our point-by-point responses to each of the reviewers’ comments are set out in the attached and changes to the manuscript are highlighted in yellow. We hope that you now find the manuscript suitable for publication.
Please see the attachment.
Thank you for your consideration. I look forward to hearing from you.

Reviewer 2 Report
The authors have expanded on their previous studies by studying the sufficiency of MF to reduce neuroinflammation. The set of experiments and data support this role. I only have a minor comment. It is unclear how the authors came to the dosing of MF vs. their previous studies using Fructus mume. Do these doses produce similar MF activity compared to their previous studies? Is it more? A clear justification would benefit the connection to the previous studies.
Author Response

(The authors gave the same response as above.)
